# How the COVID-19 Pandemic Has Changed Adolescent Health: Physical Activity, Sleep, Obesity, and Mental Health

**DOI:** 10.3390/ijerph19159224

**Published:** 2022-07-28

**Authors:** Na-Hye Kim, Jung-Min Lee, Eunhye Yoo

**Affiliations:** 1Department of Taekwondo, Kyung Hee University, Global Campus, 1732 Deokyoungdaero, Giheung-gu, Yongin-si 17014, Gyeonggi-do, Korea; na819@khu.ac.kr; 2Department of Physical Education, Kyung Hee University, Global Campus, 1732 Deokyoungdaero, Giheung-gu, Yongin-si 17014, Gyeonggi-do, Korea; jungminlee@khu.ac.kr; 3Sports Science Research Center, Kyung Hee University, Global Campus, 1732 Deokyoungdaero, Giheung-gu, Yongin-si 17014, Gyeonggi-do, Korea; 4Department of Physical Education, Seoul National University, 1 Gwanakro, Gwanakgu, Seoul 08826, Korea

**Keywords:** COVID-19, adolescent health, physical activity, sleep, obesity, mental health

## Abstract

The purpose of this study is to provide essential data for the establishment of education and policy for the formation of healthy lifestyles of adolescents in the future by analyzing the patterns of changes in society due to the prolonged COVID-19 in the physical activities, sleeping habits, obesity, and mental health of Korean adolescents. To this end, a total of 147,346 adolescents were selected and analyzed according to the purpose of the study in the 2018 (14th), 2019 (15th), and 2020 (16th) raw data of the “Youth Health Behavior Online Survey,” an annual national approval statistical survey conducted by a Korean government agency. The study examined changes in the physical activity, obesity, sleep, and mental health of Korean adolescents due to COVID-19. The physical activity rate of Korean adolescents in 2019 decreased by 5.3% from 2018. In addition, the physical activity rate in 2020 decreased by 2.1% compared to 2019. It was found that physical activity steadily decreased (*p* < 0.001). The obesity rate increased by 0.9% in 2019 compared to 2018 and by 1.8% in 2020 compared to 2019. Although the obesity rate steadily increased, it was found that it was accelerated due to COVID-19 (*p* < 0.001). Looking at the subjective sleep satisfaction rate of Korean adolescents, in 2019, it was 0.1% lower than in 2018, while in 2020, when COVID-19 began, it increased by 3.5% compared to 2019. It was found that satisfaction with sleep increased after COVID-19. Finally, the mental health characteristics of Korean adolescents by year were divided into stress and depression. Stress decreased by 1% compared to 2019 and 2018 and by 6.2% compared to 2020 and 2019. Depression increased by 1% in 2019 compared to 2018 and decreased by 3.4% in 2020 compared to 2019. In other words, stress and depression decreased after COVID-19. In 2020, when COVID-19 occurred, it was confirmed that there was a change in the health behavior of adolescents compared to 2018 and 2019. Therefore, active responses from schools, families, and communities are required to foster healthy lifestyle habits in social changes such as COVID-19.

## 1. Introduction

Coronavirus disease 2019 (COVID-19) is a novel infectious disease that was first reported in Hubei Province, China in December 2019 [1]. COVID-19 has spread worldwide since January 2020, and as of 11 March 2020, the World Health Organization (WHO) has declared a pandemic, the highest risk level in the WHO epidemic alert level [1]. To prevent the spread of COVID-19, preventive measures such as wearing masks and social distancing were exercised and have subsequently brought fundamental changes in people’s daily lives [2]. Non-face-to-face “untact” has become the norm for individuals and society as a whole [3]. The fear of COVID-19 and policies to control public activities has led to decreased social activities and weakened the vitality of societies [4]. In particular, the COVID-19 pandemic has had substantial effects on education. Schools were first closed down in China in February 2020, and by the end of June 2020, the Organization for Economic Co-operation and Development (OECD) member countries and partner countries closed down schools for 7 to 19 weeks [5]. More than 1.576 billion students in 18 countries were unable to attend school and instead participated in online classes [6]. In Korea, the first case of COVID-19 infection was reported on 20 January 2020, and the start of the school semesters in March was postponed due to the nationwide spread of the disease. For the first time in history, on 9 April 2020, online classes were conducted for third-year middle and high school students. The physical attendance of students has changed according to the number of confirmed COVID-19 cases. Due to the prolonged nature of the COVID-19 pandemic and to prevent the spread of the disease, non-routine school attendance such as biweekly and alternating day school systems have been implemented for each grade.

The home and school are the key social and environmental spaces that affect adolescent health [7]. Health behaviors and related lifestyles acquired during adolescence from the age of 12 to 18 continue through adolescence, middle age, and late-middle age into old age [8,9,10], suggesting that health behaviors in adolescence not only affect adolescent growth, but also social health. In particular, the peer group in adolescence provides important social and emotional support and has significant effects on the socialization of various behaviors. Thus, school life provides an important developmental environment for the cultivation of healthy habits in adolescents. However, the closure of schools and changes in the classroom environment due to COVID-19 has restricted systematic physical activities and increased sedentary time, threatening the quality of life of adolescents. According to the 2020 Youth Comprehensive Survey by the Ministry of Gender Equality and Family [11], adolescents responded that negative changes in school life, trust in society, career, and employment prospects greatly outweighed positive changes due to COVID-19, and 46% of adolescents experienced increased stress. These unprecedented changes in education and living environment due to COVID-19 led to decreased physical activity and strength, changes in sleep patterns, and decreased opportunities for social activities, and thereby had negative effects on the healthy lifestyle of adolescents [12,13,14].

Different measures are needed to protect the healthy lifestyle and psychological health of adolescents during the COVID-19-driven social isolation [15]; however, there are difficulties in seeking such measures when the whole society is focused on preventing the spread of COVID-19. Additionally, adolescents in Korea had already exhibited a poor health lifestyle before the COVID-19 pandemic. The strong emphasis on academic performance in Korea has led to a lack of sleep, unbalanced nutritional intake due to irregular eating habits, decreased physical activity, and academic stress [16]. Among Korean adolescents, 68.9% of middle school students and 80.0% of high school students are stressed by academic problems [17], and depression and academic grades are cited as the main causes of youth suicide [18,19,20]. The majority of adolescence is spent learning and studying [4], and approximately 95% of Korean adolescents do not follow the WHO’s recommendation of engaging in moderate or higher intensity physical activity for an average of 60 min per day [21]. About 5% of Korean adolescents meet the level of recommended physical activity of WHO, which is very low, and the rate of participation in physical activity is very low compared to that of American adolescents [22]. Adolescents in Korea suffer from a serious amount of sleep deprivation and have the lowest sleep duration when compared to adolescents in the U.S., Japan, and China [16]. Moreover, in Korea, 25.2% of adolescents experience depression, and suicide has been the leading cause of death in adolescents for many years [23]. Thus, it is difficult for Korean adolescents to cultivate healthy habits and lifestyles.

A healthy lifestyle formed in adolescence is a key factor that enhances the quality of life throughout adulthood. As the COVID-19 pandemic is continuing, the possibility of life with COVID-19 (“With COVID-19”) has been suggested, and it is important to assess what kind of lifestyle adolescents are forming in the midst of the social and environmental upheaval caused by COVID-19. Therefore, considering the cultural characteristics of Korean society, this study analyzed differences in the health behavior of adolescents before and after COVID-19 to provide basic data for education and policies on adolescent health behavior.

## 2. Materials and Methods

### 2.1. Research Participants

Raw data from the 14th (2018), 15th (2019), and 16th (2020) Adolescent Health Behavior Survey were used in this study. The Adolescent Health Behavior Survey is a nationally approved statistical survey that has been conducted annually since 2005 to determine the status and level of the health behaviors of Korean adolescents based on the National Health Promotion Act (Article 19) by the Korea Disease Control and Prevention Agency, Ministry of Health and Welfare of South Korea, and Ministry of Education, Science, and Technology (approval number 117058). Data from middle and high schools nationwide as of April 2019 and August 2020 were used. Sampling consisted of three stages: population stratification, allocation, and sampling. Regional groups and school levels were used as stratification variables in the population stratification stage. In the sample distribution stage, the proportional distribution method was applied to distribute it so that the population composition ratio and the sample composition ratio for each stratified variable match. For sampling, a stratified colony extraction method was used. Moreover, the primary extraction unit was a school, and the second extraction unit was a class.

In the process of data collection, the teacher at the sample school helping with the survey explained the outline and method of the study to students. In accordance with the guidelines of the Korea Disease Control and Prevention Agency, the teacher explained the purpose of the survey to the sample class and told them how they could participate in it. The survey was conducted for 45–50 min in a computer room with Internet access using an anonymous self-reporting method. The 14th Adolescent Health Behavior Survey was conducted in April 2018 on 62,823 adolescents, and a total of 60,040 adolescents from 800 schools participated in the survey (participation rate 95.6%). The 15th Adolescent Health Behavior Survey was conducted in April 2019 on 60,100 adolescents, and a total of 57,303 adolescents from 800 schools participated (participation rate 95.3%). On the other hand, due to COVID-19, the 16th Adolescent Health Behavior Survey was conducted from May to November 2020 on 57,925 adolescents, and a total of 54,948 adolescents from 793 schools participated (participation rate 94.9%). The raw data were acquired from the Adolescent Health Behavior Online Survey Internet website by the authors. According to the purpose of this study, 24,945 adolescents who did not respond to some of the items were excluded, and a total of 147,346 adolescents from the first year of middle school to the third year of high school were included in the final analysis. This work was supported by the Ministry of Education of the Republic of Korea and the National Research Foundation of Korea (NRF-2021S1A5A8064404).

### 2.2. Evaluation Items and Methods

In this study, the general characteristics and health characteristics were analyzed. The general characteristics included sex, age, school level, residential area, academic performance, economic level, height, weight, and BMI. The health characteristics included the physical activity rate, obesity, sleep, and mental health. The details are described as follows.

#### 2.2.1. Physical Activity

The physical activity rate was assessed using raw data on “days of physical activity for more than 60 min”, “days of intense physical activity”, and “days of muscle strengthening exercise”. According to the WHO Recommended Guidelines for Physical Activity, the physical activity rate was evaluated as “very insufficient” for not practicing any of the three types of physical activity, “insufficient” for practicing 1–2 types of activities, and “sufficient” for practicing all types of physical activity [24].

#### 2.2.2. Obesity

Obesity was assessed using BMI, the subjective perception of body type, and efforts made for weight control. The BMI < 5th percentile, 5th percentile ≤ BMI < 85th percentile, 85th percentile ≤ BMI 95th percentile, and BMI ≥ 95th percentile were considered underweight, normal weight, overweight, and obese, respectively. Subjective perception of body type was originally assessed into five categories from “very skinny” to “very fat” using the following item: “What is your opinion on your body type?”. In this study, subjective perception of body type was reorganized into three categories: skinny, normal, and chubby. Effort for weight control was assessed using the item “Have you tried to control your weight in the last 30 days?” and as specified in the original data, responses were classified into “I do not manage my weight”, “I try to lose weight”, “I try to gain weight”, and “I try to maintain weight”.

#### 2.2.3. Sleep

Sleep was assessed based on the average daily sleep duration, satisfaction of recommended sleep duration, and subjective sleep duration satisfaction rate. The average daily sleep duration referred to the mean number of hours of sleep from Monday to Friday. Satisfaction of recommended sleep duration was assessed as “insufficient,” “sufficient,” and “excessive” if the average daily sleep duration was less than seven hours, between seven and nine hours, and more than nine hours, respectively. The subjective sleep satisfaction rate was assessed using the item “Have you slept enough in the past seven days to recover from fatigue?” and as specified in the original data, responses were classified into “very sufficient”, “sufficient”, “neutral”, “insufficient”, and “very insufficient”.

#### 2.2.4. Mental Health Management

Mental health was assessed based on the level of stress and depression. In the original data, stress was evaluated using the item “How much stress do you feel in general?” and responses were classified into five grades from “highly stressed” to “not stressed at all”. In this study, stress variables were reclassified into three categories based on prior research [25,26,27], “high stress”, “a little stress”, and “no stress at all.” Depression was evaluated using the item “Have you felt sad or hopeless that you stopped your daily life for two weeks in the last 12 months?” and as specified in the original data, responses were classified into “yes” and “no”.

### 2.3. Data Analysis

Data were analyzed using the SPSS 25.0 (SPSS Inc. Chicago, IL, USA) program. Descriptive statistics were conducted for the frequency analysis of the general characteristics, and Chi-square test (χ^2^ test) was conducted to analyze differences in the health habits before and after the COVID-19 pandemic. A *p*-value of < 0.05 was considered statistically significant.

## 3. Results

### 3.1. General Characteristic of the Subjects

To analyze the general characteristics of adolescents, the frequency and percentage of age, school level, residential area, academic performance, economic level, and physical characteristics were compared according to sex. A total of 147,346 adolescents from the first year of middle school to the third year of high school were included in the analysis, and among them, 38,123 (50.6%), 36,697 (51.0%), 37,222 (49.4%), and 35,404 (49.0%) were male middle school students, female middle school students, male high school students, and female high school students, respectively. The proportion of students with high academic performance was higher among male students (40.1%) than among female students (37.4%). The proportion of those with an average academic performance was lower in male students (28.9%) than in female students (31.3%), and the proportion of students with low academic performance was smaller among male students (31.0%) than among female students (31.3%). The proportion of male students at a high economic level (42.2%) was higher than that of female students (37.0%). The proportion of those at a medium economic level was lower in male students (45.3%) than in female students (50.0%). Additionally, the proportion of male students at a low economic level (12.4%) was lower than that of the female students (13.0%).

The mean weight of adolescents involved in the study was 59.08 ± 12.88 (kg), and the mean BMI was 21.38 ± 3.55 (kg·m^2^). According to gender, the mean height of male students was 170.67 ± 7.87 (cm) and the mean weight was 64.49 ± 13.71 (kg), with a BMI of 22.02 ± 3.87 (kg·m^2^). The mean height of female students was 160.41 ± 5.43 (cm) and the mean weight was 53.42 ± 8.94 (kg), with an mean BMI of 20.72 ± 3.04 (kg·m^2^). The results of the frequency analysis of age, education, location, academic grades, economic level of participants according to gender are shown Table 1.

### 3.2. Differences in Health Habits of Korean Adolescents before and after COVID-19 per Year

#### 3.2.1. Physical Activity Rate

There were significant differences in the physical activity rate for each year before and after COVID-19 (*p* < 0.001). Most students did not perform physical activities (very insufficient, 59.7%). In 2020, 63.0% of the participants did not perform any physical activities, which was higher than the percentage in 2019 (60.9%) and 2018 (55.6%). The proportion of total students who performed “1–2 types” of physical activity in 2020 was 32.4%, which was the lowest compared to that in 2019 (34.6%) and 2018 (40.5%). Additionally, 4.6% of students performed all “three types” of physical activity recommended by the WHO in 2020, which was higher than that in 2019 (4.4%) and 2018 (3.9%).

#### 3.2.2. Obesity Characteristics

There were significant differences in the obesity characteristics of Korean adolescents for each year before and after COVID-19 (*p* < 0.001). The highest proportion of total students (50.3%) had a normal weight according to BMI, and the highest subjective perception of body type was chubby at 37.9%. In addition, the greatest proportion of total students (46.8%) did not manage their weight. The proportion of underweight students was 20.9% in 2020, which was lower than that in 2019 (21.6%) and 2018 (21.4%), and the proportion of normal weight was 48.5% in 2020, which was lower than that in 2019 (50.5%) and 2018 (51.6%). The proportion of overweight students increased in 2020 to 13.5% compared to that in 2019 (12.5%) and 2018 (12.5%). The rate of obesity was 17.2% in 2020, which was higher than that in 2019 (15.4%) and 2018 (14.5%). Regarding the subjective perception of body type, the rate of skinny students was 24.5% in 2020, which was lower than that in 2019 (25.7%) and 2018 (25.4%). The proportion of chubby students was 38.6% in 2020, which was higher compared to 2019 (37.5%) and 2018 (37.3%). The proportion of students who did not manage their weight was 45.4% in 2020, which was lower than that in 2019 (47.5%) and 2018 (47.3%). The proportion of students who tried to lose weight was 34.5% in 2020, which was higher compared to 2019 (33.0%) and 2018 (33.7%). The proportion of students who tried to gain weight was 7.7% in 2020, which was higher than that in 2019 (7.4%) and 2018 (6.8%). Additionally, 12.4% of students tried to maintain their weight in 2020, which was greater than the percentages in 2019 (12.0%) and 2018 (12.2%). 

#### 3.2.3. Sleep Characteristics

There were significant differences in the sleep characteristics of Korean adolescents for each year before and after COVID-19 (*p* < 0.001). The average daily sleep duration of total students was 6.99 ± 1.52 h, and the majority of students (50.9%) did not meet the recommended hours of sleep per day. For the subjective rate of sleep satisfaction, the proportion of students with neutral satisfaction was the greatest at 33.6%. The average daily sleep duration was 6.94 ± 1.50 h in 2020, which was lower than that in 2019 (7.02 ± 1.52 h) and 2018 (7.00 ± 1.54 h). The proportion of students who did not meet the recommended hours of sleep for adolescents was 52.1% in 2020, which was higher than that in 2019 (49.9%) and 2018 (50.9%). Those who had “sufficient” hours of sleep accounted for 39.6% in 2020, which was lower compared to 2019 (41.1%) and 2018 (40.0%). Additionally, the proportion of students with “excessive” hours of sleep was 8.3% in 2020, which was lower than that in 2019 (9.0%) and 2018 (9.1%). The proportion of students with “Highly sufficient” subjective sleep satisfaction was 10.0% in 2020, which is higher compared to 2019 (6.5%) and 2018 (6.6%). The proportion of students with “neutral” subjective sleep satisfaction was 34.5% in 2020, which was higher than that in 2019 (32.7%) and 2018 (33.7%). Finally, the proportion of students with “Highly insufficient” subjective sleep satisfaction was 9.7% in 2020, which was lower than that in 2019 (15.7%) and 2018 (13.0%).

#### 3.2.4. Mental Health Characteristics

There were significant differences in the mental health characteristics of Korean adolescents for each year before and after COVID-19 (*p* < 0.001). Most students (42.5%) faced a “moderate” level of stress, and the majority (73.9%) did not experience depression. The proportion of students with “high” levels of stress was 33.1% in 2020, which was lower compared to 2019 (39.3%) and 2018 (40.3%). The proportion of students with “moderate” levels of stress was 45.0% in 2020, which had increased compared to 2019 (41.3%) and 2018 (41.3%). Additionally, the proportion of students with “low” levels of stress was 21.9% in 2020, which was higher than that in 2019 (19.4%) and 2018 (18.3%). The proportion of students who did not have experiences of depression was 75.9% in 2020, which was higher than that in 2019 (72.5%) and 2018 (73.5%). The proportion of those with experiences of depression was 24.1% in 2020, which had decreased compared to that in 2019 (27.5%) and 2018 (26.5%). Table 2 examines the differences in physical activity, obesity, sleep, and mental health by year before and after COVID-19. 

## 4. Discussion

Although the rate of COVID-19 infection is low in adolescents, they are in a vulnerable developmental environment as their families, schools, and local communities are greatly affected by the pandemic. Negative experiences such as fear of infection, lack of outdoor activities, and adaptive stress lead to limited experiences, physical activities, and socialization opportunities, which are critical in the growth and development phase of life. Therefore, in this study, we analyzed differences in the lifestyle habits of adolescents, which can affect their quality of life throughout their lives, before and after the COVID-19 pandemic. We observed that significant changes had taken place in the physical activity rate, obesity characteristics, sleep characteristics, and mental health characteristics of adolescents after the COVID-19 pandemic.

The level of physical activity decreased in 2020 when the COVID-19 pandemic was declared compared to that in 2019 and 2018 (*p* < 0.001). In a study conducted in 2016 by WHO on the physical activity of male and female students between the ages of 11 and 17 in 146 countries around the world, a total of 81.1% of the students did not meet the level of physical activity recommended by the WHO. In Korea, 94.2% of the students did not meet the physical activity recommendations, which was the highest rate among all countries analyzed in that study. Compared to other countries, Korea has a high national income and insufficient level of physical activities for adolescents. This may be attributed to the highly competitive academic environment. High intensity physical activities are mostly practiced during physical education classes in schools and as part of school sporting club activities [28]. To improve and substantiate the participation of adolescents in sports activities, policies are being designed to provide a conducive environment and sufficient time for participation [11]. The level of physical activity increased from 10% in 2010 to 14% in 2020 [29]; however, the closure of schools due to the spread of COVID-19 has restricted opportunities for after-school physical activities, and the lack of outdoor activities due to social distancing has decreased the level of physical activity.

This study showed that the proportion of students with the subjective perception of chubby body type and who tried to lose weight increased in 2020 compared to that in 2019 and 2018 (*p* < 0.001). In the last 40 years, the rate of obesity in children and adolescents worldwide has increased by ten-fold [30], and the rate of obesity in Korean adolescents is also gradually increasing. Although the rate of obesity in adults is low in Korea compared to other countries, the rate in male adolescents is higher than the OECD average [31]. Thus, the Korean government has sought National Obesity Management Comprehensive Measures (2018–2020) to reduce the rate of obesity in the community by improving eating habits and encouraging physical activities. Considering this trend of increase in obesity rates, the results of this study cannot be solely attributed to the COVID-19 pandemic. However, a study on changes in body weight during the COVID-19 pandemic in Korean adults over the age of 20 found that 46% of the participants had gained 3 kg or more due to reduced daily activities and exercise [32]. In another study, reduced physical activities, changes in dietary patterns, and increased inactive time caused seasonal differences in the normal growth pattern of Korean children and adolescents while increasing BMI [33]. These findings suggest that difficulties to maintain a healthy lifestyle during the COVID-19 pandemic have led to an increased rate of obesity. Additionally, multi-faceted school-centered interventions for the prevention of obesity in children and adolescents that involved families and local communities have been effective in preventing obesity [34,35]. However, restrictions in such interventions due to COVID-19 are likely to have affected healthy behavior and weight loss, thereby increasing the obesity rate.

The average daily sleep duration and the proportion of students who received the recommended hours of sleep decreased in 2020 compared to 2018 and 2019 (*p* < 0.001). On the other hand, the proportion of students with “sufficient” or greater subjective sleep satisfaction increased in 2020 compared to 2018 and 2019 (*p* < 0.001). Many adolescents experience sleep deprivation [36]. Prior to the COVID-19 pandemic, 53.9% of middle school students and 73% of high school students in Korea perceived that they had insufficient sleep duration [37]. Adolescents between the age of 14 and 17 require an average of eight to 10 h of sleep per day [30]; however, only 29.6% of high school students in Korea slept more than seven hours a day, which is substantially lower than the corresponding percentage (46.7%) in the U.S. [38]. However, although the average sleep duration has decreased in adolescents after the COVID-19 pandemic, satisfaction with sleep has increased. This suggests that the quality of sleep is relatively better. Sleep is a biological phenomenon affected by the complex effects of multiple social changes [39]. In adolescents, environmental factors such as excessive schoolwork and the use of electronic devices may lead to decreased sleep duration [40]. COVID-19 has caused adolescents to have difficulties in engaging in social activities such as going to school. In addition, adolescents who deviate from the minimum social management network are placed in an environment where they can freely use PCs or mobile devices, resulting in irregular life patterns that means that they fall asleep late or lack self-control. However, it is judged that adolescents are satisfied with sleep by adjusting their lifestyle patterns by their biological sleep needs rather than dissatisfaction with the lack of sleep in a relatively free life [41].

We observed that stress and depression decreased in 2020 compared to 2018 and 2019 (*p* < 0.001). Unusual experiences often lead to negative emotions such as fear, worry, and anger [42]. Previous studies have shown that people develop a fear of contact with others due to the high rate of infection and mortality by COVID-19 [43], and adolescents in their 10s and 20s are psychologically more affected by the pandemic than those in their 40s and 50s as their daily lives are restricted [44]. Based on these findings, it was expected that stress and depression in adolescents would have increased after the COVID-19 pandemic began. However, the decreased stress and depression in adolescents in this study suggest that changes in daily life caused by the pandemic have had positive effects on mental health. This can be interpreted as a temporary change in the lives of academic-oriented adolescents due to COVID-19. The COVID-19 pandemic led to reduced hours of studying and private education [23], and worry about academic performance, which is the biggest cause of stress, may have decreased. 

Adolescents experience stress and develop feelings of depression in school. Schools based on group life provide a space specific for the study and competition through education and help adolescents socialize with their peers. However, school environments lack the consideration of individual traits and levels, and this may lead to negative experiences such as school violence and feelings of exclusion from peer relationships. In particular, younger generations of adolescents are familiar with digital technologies and mobile content. These adolescents have quickly adapted to online classes that allow them to receive education in their personal space without physical contact. They are able to maintain peer relationships through Internet-based platforms such as social networks and online games while being away from the negative social and psychological experiences that often occur in a school environment. In other words, it is considered that adolescents had an environment in which they could not feel isolated by maintaining social relationships in an online space, and not in a physical space, and that factors that would negatively affect mental health did not increase due to COVID-19. Altogether, the adolescents’ physical and mental health is likely to have been relatively less affected by the COVID-19 pandemic. However, the COVID-19 pandemic has significantly increased the weekly Internet use from 17.6 h in 2019 to 27.6 h in 2020, and the proportion of those with a risk of overdependence on smartphones has increased by 5.6% to 35.8% in 2020 compared to 2019 [23]. Therefore, in establishing support policies related to adolescent mental health after COVID-19, it is necessary to study the correlation between adolescent mental health and school, and based on this, it is necessary to establish an education policy considering the characteristics of adolescents familiar with the Internet environment. 

This study is in line with the study by Loades (2020) [45], where social isolation and loneliness caused depression and anxiety in adolescents. In addition, a study showing that home learning and social distancing caused by COVID-19 affected adolescent depression, anxiety, sleep, and dietary disorders showed different results on adolescent mental health during COVID-19 [46]. However, this study used data collected in the early stages of COVID-19 and data before COVID-19, and there is a limitation in that it was not a study that derived active support or prevention policies for various mental health problems. Therefore, attention is required in interpreting and applying results, and research that can compare various information with longitudinal studies in the future needs to be conducted. In particular, to manage the mental health of COVID-19 adolescents, it is judged that various situations should be considered as mental health problems, and various problems that occur in the adaptation process may appear. 

This study is significant in that it used a large amount of publicly trusted data by nationally investigating the health behavior of adolescents and conducting a study on the subject of COVID-19, which has affected the lives of the world’s population. In addition, it was judged to be significant in that it did not simply compare the data on youth health behavior before the outbreak of COVID-19 (2019) and after the outbreak of COVID-19, but compared the 2018 data together to confirm whether it was a trend among adolescent health behavior or a change due to the outbreak of COVID-19.

On the other hand, this study had the following limitations. First, due to the complex connected social structure, the causal relationship was not derived by controlling all of the disturbance variables due to COVID-19, and second, the relationship between youth physical activity, sleep, obesity, and mental health was analyzed through cross-analysis, so there was a limit to explaining the size or predictive model of the health behavior factors. Therefore, in the follow-up study, it is necessary to study the health behavior of adolescents by region or to derive a model to predict the health behavior of adolescents in a pandemic situation such as COVID-19.

## 5. Conclusions

Adolescence is a dynamic phase of social change and new physical and mental experiences in life. In such unprecedented circumstances of high social anxiety due to COVID-19, adolescents who are in their growth and development phase of life may be vulnerable to various risks. In this paper, we indirectly suggest that the lifestyle and health habits of adolescents have changed due to COVID-19 and showed that they adapted to the pandemic in a pattern different from that of the older generation. COVID-19 has led to changes in the physical activity rate, obesity characteristics, and sleep characteristics that could harm the health of adolescents. Active responses from schools, families, and local communities are required to help adolescents cultivate healthy lifestyles. The increased subjective sleep satisfaction and decreased stress and depression observed in our study indicate that new educational methods promoting a healthy lifestyle in adolescents and a review of the changed adolescent lifestyle may be necessary, rather than simply returning back to pre-COVID-19 lifestyles.

## Figures and Tables

**Table 1 ijerph-19-09224-t001:** The general characteristics of the subjects (*n* = 147,346).

Variables		Total	Male	Female
		*n* (%)	*n* (%)	*n* (%)
Age (year)	12	10,730 (7.3%)	5344 (7.1%)	5386 (7.5%)
	13	24,870 (16.9%)	12,650 (16.8%)	12,220 (17.0%)
	14	24,877 (16.9%)	12,701 (16.9%)	12,176 (16.9%)
	15	24,579 (16.7%)	12,639 (16.8%)	11,940 (16.6%)
	16	24,112 (16.4%)	12,421 (16.5%)	11,691 (16.2%)
	17	25,011 (17.0%)	12,702 (16.9%)	12,309 (17.1%)
	18	13,167 (8.9%)	6888 (9.1%)	6279 (8.7%)
Education	Middle school	74,820 (50.8%)	38,123 (50.6%)	36,697 (51.0%)
	High school	72,526 (49.2%)	37,222 (49.4%)	35,304 (49.0%)
Location	Urban	75,141 (51.0%)	4326 (5.7%)	4460 (6.2%)
	Suburban	63,419 (43.0%)	32,345 (42.9%)	31,074 (43.2%)
	Rural	8786 (6.0%)	38,674 (51.3%)	36,467 (50.6%)
Academic grades	High	57,126 (38.8%)	30,221 (40.1%)	26,905 (37.4%)
	Medium	44,325 (30.1%)	21,770 (28.9%)	22,555 (31.3%)
	Low	45,895 (31.1%)	23,354 (31.0%)	22,541 (31.3%)
Economic level	High	58,425 (39.7%)	31,815 (42.2%)	26,610 (37.0%)
	Medium	70,185 (47.6%)	34,168 (45.3%)	36,017 (50.0%)
	Low	18,736 (12.7%)	9362 (12.4%)	9374 (13.0%)
Total		147,346 (100%)	75,345 (51.1%)	72,001 (48.9%)
Variables		Mean ± SD	Mean ± SD	Mean ± SD
Anthropometrics	Height (cm)	165.66 ± 8.51	170.67 ± 7.87	160.41 ± 5.43
	Weight (kg)	59.08 ± 12.88	64.49 ± 13.71	53.42 ± 8.94
	BMI (kg·m^2^)	21.38 ± 3.55	22.02 ± 3.87	20.72 ± 3.04

Note: The total number of study participants was 147,346, and the frequency analysis and descriptive statistics analysis were conducted to review the demographic characteristics according to their gender. Depending on gender, the values of age, education, location, academic grades, and economic level were expressed as frequency (*n*) and percentage (%), and the values of anthropometrics were presented as the mean ± standard deviation.

**Table 2 ijerph-19-09224-t002:** The differences in physical activity, obesity, sleep, and mental health by year before and after COVID-19.

Variables		Year			Total *n* (%)	χ^2^	*p*
		2018 *n* (%)	2019 *n* (%)	2020 *n* (%)			
Level of physical activity	Very insufficient	28,682 (55.6)	30,725 (60.9)	28,576 (63.0)	87,983 (59.7)	760.805	0.001
	Insufficient	20,882 (40.5)	17,478 (34.6)	14,672 (32.4)	53,032 (36.0)		
	Sufficient	1986 (3.9)	2242 (4.4)	2103 (4.6)	6331 (4.3)		
Body Mass Index (kg/m^2^)	Underweight	11,032 (21.4)	10,906 (21.6)	9441 (20.8)	31,379 (21.3)	200.907	0.001
	Normal	26,625 (51.6)	25,477 (50.5)	21,981 (48.5)	74,083 (50.3)		
	Overweight	6421 (12.5)	6305 (12.5)	6112 (13.5)	18,838 (12.8)		
	Obese	7472 (14.5)	7757 (15.4)	7817 (17.2)	23,046 (15.6)		
Subjective body shape perception	Skinny	13,112 (25.4)	12,973 (25.7)	11,116 (24.5)	37,201 (25.2)	23.196	0.001
	Average	18,999 (36.9)	18,541 (36.8)	16,729 (36.9)	54,269 (36.8)		
	Fat	19,439 (37.7)	18,931 (37.5)	17,506 (38.6)	55,876 (37.9)		
Weight control	None	24,384 (47.3)	23,978 (47.5)	20,580 (45.4)	68,942 (46.8)	71.780	0.001
	Losing	17,362 (33.7)	16,659 (33.0)	15,650 (34.5)	49,671 (33.7)		
	Maintaining	6279 (12.2)	6073 (12.0)	5636 (12.4)	17,988 (12.2)		
	Gaining	3525 (6.8)	3735 (7.4)	3485 (7.7)	10,745 (7.3)		
Average daily hours of sleep (hours)		7.00 ± 1.54	7.02 ± 1.52	6.94 ± 1.50	6.99 ± 1.52		
Whether recommended hours of sleep are attained	<7(fell short)	26,256 (50.9)	25,166 (49.9)	23,615 (52.1)	75,037 (50.9)	55.041	0.001
	≥7~>9(reached)	20,614 (40.0)	20,753 (41.1)	17,952 (39.6)	59,319 (40.3)		
	≥9(exceeded)	4680 (9.1)	4526 (9.0)	3784 (8.3)	12,990 (8.8)		
Awareness of subjective sleep satisfaction	Highly sufficient	3408 (6.6)	3259 (6.5)	4522 (10.0)	11,189 (7.6)	1945.852	0.001
	Sufficient	8542 (16.6)	7654 (15.2)	9464 (20.9)	25,660 (17.4)		
	Neutral	17,349 (33.7)	16,512 (32.7)	15,628 (34.5)	49,489 (33.6)		
	Insufficient	15,562 (30.2)	15,083 (29.9)	11,352 (25.0)	41,997 (28.5)		
	Highly insufficient	6689 (13.0)	7937 (15.7)	4385 (9.7)	19,011 (12.9)		
Stress	Low	9412 (18.3)	9769 (19.4)	9942 (21.9)	29,123 (19.8)	647.309	0.001
	Medium	21,387 (41.5)	20,852 (41.3)	20,413 (45.0)	62,652 (42.5)		
	High	20,751 (40.3)	19,824 (39.3)	14,996 (33.1)	55,571 (37.7)		
Awareness of depression experience	No	37,902 (73.5)	36,570 (72.5)	34,434 (75.9)	108,906 (73.9)	152.121	0.001
	Yes	13,648 (26.5)	13,875 (27.5)	10,917 (24.1)	38,440 (26.1)		

Note: The values are presented as the means ± standard deviations. Descriptive statistical analysis and cross-analysis were conducted to compare the differences between the variables of Korean adolescents by year. The average daily hours of sleep did not perform cross-analysis on a ratio scale, and as a result of cross-analysis on all remaining variables, the derived *p*-value was found to be statistically significant (*p* < 0.000).

## Data Availability

The datasets used and/or analyzed during the current study are available from the corresponding author on reasonable request.

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
