# Peer review of "How the COVID-19 Pandemic Has Changed Adolescent Health: Physical Activity, Sleep, Obesity, and Mental Health"

_ijerph, 2022, doi:10.3390/ijerph19159224_

Round 1
Reviewer 1 Report
I wish to thank the authors for their work conducting this study and preparing a well-organized and informative manuscript. This paper is highly readable and very interesting due to the large volume of participants in South Korea. I thoroughly enjoyed reading this paper.
The specific concerns that I have with the manuscript are listed below:
The introduction clearly stated the purpose and need for this study. Since the authors measured a variety of factors, it would be useful to provide additional information in the introduction about the context of youth obesity in South Korea. It can be related to the physical activity component (e.g., in lines 66-70).
Lines 46-53: The authors should consider including relevant citations/resources to support the background.
Lines 55-57: The authors should include the specific range of age groups, including the age range of South Korean adolescents.
Lines 75-78: The authors could consider including additional information regarding South Korea's cultural/contextual backgrounds (e.g., competitive educational system) in their description of the link between the research variables. Also, it would also be beneficial to utilize the more recent citations to support the sentence.
Lines 79-81: The authors might consider including quantitative data on the status of youth physical activity in South Korea.
Lines 290-304: The authors should explicitly highlight the research results relating to physical activity. For example, the authors may discuss the characteristics of youth daily physical activity in the COVID-19 era (i.e., based on the study purpose). The present draft is mostly focused on the conditions prior to the pandemic (or without a pandemic). I would like to propose looking into school policy and/or government policy to handle environmental issues related to the pandemic (limited opportunity for participation in the community due to lack of accessible facilities, etc.).
Lines 348-366: The authors should consider elaborating on this paragraph. Previous empirical research has shown that there are positive relationships between youth physical activity participation and psychosocial (mental health) outcomes.
For instance,
Biddle, S. J., Ciaccioni, S., Thomas, G., & Vergeer, I. (2019). Physical activity and mental health in children and adolescents: An updated review of reviews and an analysis of causality. Psychology of Sport and Exercise, 42, 146-155.
Dale, L. P., Vanderloo, L., Moore, S., & Faulkner, G. (2019). Physical activity and depression, anxiety, and self-esteem in children and youth: An umbrella systematic review. Mental Health and Physical Activity, 16, 66-79.
Rodriguez-Ayllon, M., Cadenas-Sánchez, C., Estévez-López, F., Muñoz, N. E., Mora-Gonzalez, J., Migueles, J. H., ... & Esteban-Cornejo, I. (2019). Role of physical activity and sedentary behavior in the mental health of preschoolers, children and adolescents: a systematic review and meta-analysis. Sports medicine, 49(9), 1383-1410.
One of the intriguing findings from the current study is that youth stress and depression (i.e., mental health) may be reduced despite restricted opportunities for physical activity involvement and socialization during COVID-19 (i.e., unhealthy lifestyle context). The authors should describe the discussion more thoroughly in order to emphasize the outcomes. Expectations for high academic accomplishment might be one of the key factors influencing the outcomes; nonetheless, it should address the necessity of further study on it (with specific directions).
Lines 356-357: The authors should rephrase this sentence because of the feature of the current study (i.e., descriptive study). The study's findings do not establish a cause-and-effect link.
Lines 368-369: This statement should be readdressed or deleted by the authors. What does "MZ generation" imply, in particular? What is the connection between the variables? It should be able to accommodate further information.
Lines 371-375: The authors should address the details of the directions of the school system and policy after the pandemic (takeaway message) clearly from the findings. The authors should also mention the current study's limitations.
Author Response
Responses to Editor and Reviewer #1's Comments
We would like to thank the reviewers for the insightful comments and constructive criticism of the manuscript. We have addressed each of the specific concerns raised by the reviewers in the order that they were discussed in the comments. The requested changes have been made and designated in the red italicized text to facilitate the review. Line numbers are also provided for some responses (the line numbering restarts each page).
- Lines 55-57: The authors should include a specific range of age groups, including the age range of South Korean adolescents.
à Thanks for your suggestions. We agree with your suggestions and the sentence has been revised.
“Health behaviors and related lifestyles acquired during adolescence from the age of 12 to 18 continue through adolescence, middle age, and late middle age into old age “
- Lines 75-78: The authors could consider including additional information regarding South Korea's cultural/contextual backgrounds (e.g., competitive educational system) in their description of the link between the research variables. Also, it would also be beneficial to utilize the more recent citations to support the sentence.
à Thanks for your suggestions. We agree with your suggestions and added the following sentence.
“Among Korean adolescents, 68.9% of middle school students and 80.0% of high school students are stressed by academic problems [17], and depression and academic grades are cited as the main causes of youth suicide [18], [19], [20].”
- Lines 79-81: The authors might consider including quantitative data on the status of youth physical activity in South Korea.
à Thanks for your suggestions. We agree with your suggestions and added the following sentence.
“About 5% of Korean adolescents meet the level of recommended physical activity of WHO, which is very low, and the rate of participation in physical activity is very low compared to that of American adolescents [22].”
- Lines 290-304: The authors should explicitly highlight the research results relating to physical activity. For example, the authors may discuss the characteristics of youth daily physical activity in the COVID-19 era (i.e., based on the study purpose). The present draft is mostly focused on the conditions prior to the pandemic (or without a pandemic). I would like to propose looking into school policy and/or government policy to handle environmental issues related to the pandemic (limited opportunity for participation in the community due to lack of accessible facilities, etc.).
à Thank you for your thoughtful advice. We added to the discussion that this study is meaningful because Korea's school and government policies to deal with pandemic-related environmental issues are currently very vulnerable.
- Lines 348-366: The authors should consider elaborating on this paragraph. Previous empirical research has shown that there are positive relationships between youth physical activity participation and psychosocial (mental health) outcomes. For instance, Biddle, S. J., Ciaccioni, S., Thomas, G., & Vergeer, I. (2019). Physical activity and mental health in children and adolescents: An updated review of reviews and an analysis of causality. Psychology of Sport and Exercise, 42, 146-155. Dale, L. P., Vanderloo, L., Moore, S., & Faulkner, G. (2019). Physical activity and depression, anxiety, and self-esteem in children and youth: An umbrella systematic review. Mental Health and Physical Activity, 16, 66-79. Rodriguez-Ayllon, M., Cadenas-Sánchez, C., Estévez-López, F., Muñoz, N. E., Mora-Gonzalez, J., Migueles, J. H., ... & Esteban-Cornejo, I. (2019). Role of physical activity and sedentary behavior in the mental health of preschoolers, children and adolescents: a systematic review and meta-analysis. Sports medicine, 49(9), 1383-1410. One of the intriguing findings from the current study is that youth stress and depression (i.e., mental health) may be reduced despite restricted opportunities for physical activity involvement and socialization during COVID-19 (i.e., unhealthy lifestyle context). The authors should describe the discussion more thoroughly in order to emphasize the outcomes. Expectations for high academic accomplishment might be one of the key factors influencing the outcomes; nonetheless, it should address the necessity of further study on it (with specific directions).
à Thanks for your suggestions. We agree with your suggestions and added the following sentence.
“This study is in line with Loades (2020) [45] studies that social isolation and loneliness cause depression and anxiety in adolescents. In addition, a study showing that home learning and social distancing caused by COVID-19 affect adolescents' depression, anxiety, sleep, and dietary disorders shows different results on adolescents' mental health during COVID-19 [46]. However, this study used data collected in the early stages of COVID-19 and data before COVID-19, and there is a limitation that it is not a study that derives active support or prevention policies for various mental health problems. Therefore, attention is required in interpreting and applying results, and research that can compare various information with longitudinal studies in the future needs to be conducted. In particular, to manage the mental health of COVID-19 adolescents, it is judged that various situations should be considered mental health problems, and various problems that occur in the adaptation process may appear.”
- Lines 356-357: The authors should rephrase this sentence because of the feature of the current study (i.e., descriptive study). The study's findings do not establish a cause-and-effect link.
à Thanks for your suggestions. We agree with your suggestions and deleted and revised the following sentence.
“This study is in line with Loades (2020) [45] studies that social isolation and loneliness cause depression and anxiety in adolescents. In addition, a study showing that home learning and social distancing caused by COVID-19 affect adolescents' depression, anxiety, sleep, and dietary disorders shows different results on adolescents' mental health during COVID-19 [46].”
- Lines 368-369: This statement should be readdressed or deleted by the authors. What does "MZ generation" imply, in particular? What is the connection between the variables? It should be able to accommodate further information.
à Thanks for your suggestions. We agree with your suggestions and the sentence is reversed.
“In particular, the younger generations of adolescents are familiar with digital technologies and mobile content.”
“In other words, there was an environment where teenagers could not feel isolated by maintaining social relationships in online space rather than in physical space, and it is interpreted that factors that will have a negative impact on mental health have not increased due to COVID-19.”
- Lines 371-375: The authors should address the details of the directions of the school system and policy after the pandemic (takeaway message) clearly from the findings. The authors should also mention the current study's limitations.
à Thanks for your suggestions. We agree with your suggestions and added the following sentence.
“Therefore, in establishing support policies related to the mental health of youth after COVID-19, it is necessary to study the correlation between youth mental health and schools or studies, and based on this, it is necessary to establish educational policies taking into account the characteristics of the youth generation familiar with the Internet environment.”
“This study utilizes data collected from early COVID-19 and pre-COVID-19 data, and has limitations in that it is not a study to derive active support or preventive policies for various mental health problems. Therefore, caution is required in the interpretation and application of results, and research to compare various information with longitudinal research needs to be carried out in the future. In particular, in order to manage the mental health of COVID-19 teenagers, mental health problems and various problems that occur in the process of adaptation may occur, so various situations should be considered together.”

Reviewer 2 Report
This study only offered very simple descriptive results to present some of the conditions of physical activity, obesity, sleep, and mental health. The authors did not use appropriate statistical methods to investigate the research questions. No confounding effects have been taken into account in the analysis and interpretation for the main research questions. This study provides only fragmentary and incomplete measures of healthy lifestyle factors for adolescents.
The comments are shown in the follows.
1. Abstract. All of the study results were presented in words, no real findings were presented, i.e., there were no numerical results were given to support the arguments raised by the authors.
2. Line 26-28. In the conclusion, “adolescents need to establish a flexible and efficient educational environment”. How adolescents can themselves establish an educational environment? It is suggested to revise the statements for the conclusion.’
3. Line 43. Please describe the fully name for “OECD”.
4. Lines 54-55 and 55-56. The statement is unclear. “Home and school environments are … environmental factors”?
5. Lines 97-116. The source population was not defined and the sampling method did not be described in the manuscript.
6. Lines 138-139. Was the BMI value actually measured or self-reported by the students?
7. Lines 160-191. The descriptions of the general characteristics for the participants are too cumbersome and did not be focused.
8. Lines 192-199. The descriptions of age, weight, and BMI for the participants are too cumbersome and did not be focused.
9. Table 1: no information on any statistical tests was provided.
10. Table 2: Chi-squared tests were not an appropriate statistical method to investigate the study questions the authors raised. We are just known the association between years and variables, but are unaware which years had a higher proportion in the study variables.
11. Table 2: No confounding effects have been taken into account in the analysis and interpretation for the main research questions.
12. Lines 412-464. The displayed references 1, 5, 6, 11, 17, 18, and 24 are incomplete.
Author Response
Responses to Editor and Reviewer #2's Comments
We would like to thank the reviewers for the insightful comments and constructive criticism of the manuscript. We have addressed each of the specific concerns raised by the reviewers in the order that they were discussed in the comments. The requested changes have been made and designated in the red italicized text to facilitate the review. Line numbers are also provided for some responses (the line numbering restarts each page).
- All of the study results were presented in words, no real findings were presented, i.e., there were no numerical results were given to support the arguments raised by the authors.
- Thanks for your suggestions. We revamped the abstract.
“Abstract: The purpose of this study is to provide essential data for education and policy establishment for the formation of healthy lifestyles of adolescents in the future by analyzing the patterns of changes in society due to the prolonged COVID-19 in Korean adolescents' physical activities, sleeping habits, obesity, and mental health. To this end, a total of 147,346 adolescents were selected and analyzed according to the purpose of the study in the 2018 (14th), 2019 (15th), and 2020 (16th) raw data of the "Youth Health Behavior Online Survey," an annual national approval statistical survey conducted by a Korean government agency. The study examined changes in physical activity, obesity, sleep, and mental health of Korean adolescents due to COVID-19. The physical activity rate of Korean adolescents in 2019 decreased by 5.3% from 2018. In addition, the physical activity rate in 2020 decreased by 2.1% compared to 2019. It was found that physical activity steadily decreased (p < .001). The obesity rate increased by 0.9% in 2019 compared to 2018 and by 1.8% in 2020 compared to 2019. Although the obesity rate steadily increased, it was found that it was accelerated due to COVID-19 (p < .001). Looking at the subjective sleep satisfaction rate of Korean adolescents, 2019 was 0.1% lower than 2018, while 2020, when COVID-19 began, increased by 3.5% compared to 2019. It was found that satisfaction with sleep increased after COVID-19. Lastly, the mental health characteristics of Korean adolescents by year were divided into stress and depression. Stress decreased by 1% compared to 2019 and 2018 and by 6.2% compared to 2020 and 2019. Depression increased by 1% in 2019 compared to 2018 and decreased by 3.4% in 2020 compared to 2019. In other words, stress and depression decreased after COVID-19. COVID-19 has led to changes in physical activity rate, obesity characteristics, and sleep characteristics that could harm the health of adolescents. Active responses from schools, families, and local communities are required to help adolescents cultivate healthy lifestyles.”
- Line 26-28. In the conclusion, “adolescents need to establish a flexible and efficient educational environment”. How adolescents can themselves establish an educational environment? It is suggested to revise the statements for the conclusion.’
- Thank you for your advice. We revised the final statement
“COVID-19 has led to changes in physical activity rate, obesity characteristics, and sleep characteristics that could harm the health of adolescents. Active responses from schools, families, and local communities are required to help adolescents cultivate healthy lifestyles.”
- Line 43. Please describe the fully name for “OECD”.
- Thanks for your suggestions. We r
“Schools were first closed down in China in February 2020, and by the end of June 2020, Organization for Economic Co-operation and Development(OECD) member countries and partner countries closed down schools for 7 to 19 weeks.”
- Lines 54-55 and 55-56. The statement is unclear. “Home and school environments are … environmental factors”?
- Thanks for your suggestions. We r
“Home and school are key social and environmental spaces that affect adolescent health.”
- Lines 97-116. The source population was not defined and the sampling method did not be described in the manuscript.
- Thanks for your suggestions. We agree with your suggestions and added the following sentence.
“As a middle and high school student nationwide, the extraction framework for sample design used data from middle and high schools nationwide as of April 2018, April 2019, and August 2020. Sample extraction consisted of three stages: population stratification, sample allocation, and sample extraction. In the population stratification stage, local groups and school levels were used as stratification variables. In the sample allocation stage, the proportional allocation method was applied to ensure that the population composition ratio and the sample composition ratio by stratification variable number matched. The stratification extraction method was used for sample extraction, and the first extraction unit was school and the second extraction unit was class.”
- Lines 138-139. Was the BMI value actually measured or self-reported by the students?
- Thanks for your suggestions. We agree with your suggestions and added the following sentence.
“Students wrote their height, weight, and age independently, and the researcher calculated and classified BMI based on the original data.”
- Lines 160-191. The descriptions of the general characteristics for the participants are too cumbersome and did not be focused.
- Thanks for your suggestions. We r
“How much stress do you feel in general?” and responses were classified into five grades from “highly stressed” to “not stressed at all”. In this study, the responses were classified into three categories: “highly stressed,” “slightly stressed,” and “not stressed at all.”
- Lines 192-199. The descriptions of age, weight and BMI for the participants are too cumbersome and did not be focused.
- Thanks for your suggestions. We agree with your suggestions and the sentence is reversed.
“The mean weight of adolescents involved in the study was 59.08±12.88 (kg), and the mean BMI was 21.38±3.55 (kg·m2). According to gender, the mean height of male students is 170.67±7.87 (cm) and the mean weight is 64.49±13.71 (kg), with a BMI of 22.02±3.87 (kg·m2). The mean height of female students was 160.41±5.43 (cm) and the mean weight was 53.42±8.94 (kg), with a mean BMI of 20.72±3.04 ((kg·m2). All physical characteristics were higher for male students than for female students, and the general characteristics of other research participants were shown in <Table 1>.”
- Table 1: no information on any statistical tests was provided.
- Thanks for your suggestions. We agree with your suggestions and added the following sentence.
“The results of the frequency analysis of age, Education, Location, Academic grades, and Economic level of participants according to gender are shown <Table 1>.”
- Table 2: Chi-squared tests were not an appropriate statistical method to investigate the study questions the authors raised. We are just known the association between years and variables, but are unaware which years had a higher proportion in the study variables.
- Thanks for your comments: We totally agree with your comments
This is one of the series of manuscripts using the national data set so we first examined how adolescents’ health-related variables have proportionally changed and the second series of the manuscript that we are preparing should be examined the association between years and variables. In addition, including the association results should take a lot of information and not fit in one manuscript. Hopefully, you understood.
- Table 2: No confounding effects have been taken into account in the analysis and interpretation of the main research questions.
- Good comments.
Based on the limitation of the statistical analysis, we could not control for the confounding effects. For future research, we would like to apply multinomial logistic regression for the planning manuscript to control confounding variables.
- Lines 412-464. The displayed references 1, 5, 6, 11, 17, 18, and 24 are incomplete.
à Thanks for your suggestions. We agree with your suggestions and revised.
- WHO. Coronavirus disease (COVID-19) pandemic. Available online: https://www.who.int/emergencies/diseases/novel-coronavirus-2019 (accessed on 1 Nov 2020).
- Schleicher, A. The impact of COVID-19 on education insights from education at a glance 2020. Available online: https://www.oecd.org/education/the-impact-of-covid-19-on-education-insights-education-at-a-glance-2020.pdf (accessed on 5 Jun 2021).
- UNICEF. Education and COVID-19. Available online: https://data.unicef.org/topic/education/covid-19/ (accessed on 1 Dec 2020).
- Ministry of Gender Equality and Family, S.K. 2021 national statistics on adolescents. Available online: from http://www.kostat.go.kr (accessed on 3 Jun 2021).
- WHO. New WHO-led study says majority of adolescents worldwide are not sufficiently physically active, putting their current and future health at risk. Available online: https://iaks.sport/news/new-who-led-study-adolescents-worldwide-are-not-sufficiently-physically-active (accessed on 10 Dec 2020).
- Family, M.o.G.E.a. 2020 Comprehensive Survey on Youth. Available online: http://www.mogef.go.kr/mp/pcd/mp_pcd_s001d.do?mid=plc502&bbtSn=704797 (accessed on 21 Aug 2021).
- Obesity, K.S.f.t.S.o. A survey on the status of weight management and obesity perception in the COVID-19 era. Available online: http://general.kosso.or.kr/html/?pmode=BBBS0001300004&smode=view&seq=1372 and Adolescent Growth: Experience of Pediatric Endocrine Clinics. Accessed September 1, 2021 (accessed on 21 Aug 2021).

Reviewer 3 Report
Dear Authors,
congratulation for your article, we make some sugestion to get your article clearer for future lecturers
Necessity of the study is correctly specified in the introduction that gave a good picture of the situation in Korea.
Line 161 to 167, please find reference to justify this procedure or is it completely created by survey authors and on which methodological basis
Methodology, what are the criteriums of inclusion used to considered the subjects as adolescents? Only the age? In this case was it between 12 and 18 years for all the male and females included in this study? Is it correct or what are the limitations? You should discuss that.
Line 317 check what you written to be clearer for future lecturers.
Sleep quality improve in 2020, in the discussion you should discuss that may be because of the diminution of the circulation, less noise, less pollution, … could have had positive impact especially in the big cities. What about comparation between urban and rural on this topic?
You should discuss why you did not use statistic procedures do control confounding factors or multivariate logistic regression.
What are the limits of the study, and suggestions for future studies.
Author Response
Responses to Editor and Reviewer #3's Comments
We would like to thank the reviewers for the insightful comments and constructive criticism of the manuscript. We have addressed each of the specific concerns raised by the reviewers in the order that they were discussed in the comments. The requested changes have been made and designated in the red italicized text to facilitate the review. Line numbers are also provided for some responses (the line numbering restarts each page).
- Line 161 to 167, please find reference to justify this procedure or is it completely created by survey authors and on which methodological basis
à Thank you for your advice. The sentence was revised.
In this study, stress variables were reclassified into three categories based on prior research(Gu, 2019; Song, 2018; Kim et al., 2017): "high stress," "a little stress," and "no stress at all."
Gu, H. J. (2019). Influence of perceived stress on obesity in South Korean adolescents using data from the 13th 2017 Korea Youth Risk Behavior Web-Based Survey. Korean J Health Educ Promo, 36(1), 29-41. doi : 10.14367/kjhep.2019.36.1.29
Kim, K. J., Kim, B. S., Won, C. W., Choi, H. R., Kim, S. Y., Par, W. C., & Kwon, E. J. (2017). Relationship between adolescents health behavior, stress and birth order: The Korea youth risk behavior web-based survey 2014. The Korean Journal of Stress Research, 25(2), 138-144. doi: 10.17547/kjsr.2017.25.2.138.
Song, H. S. (2018). Gender difference in the effects of Korean youth mental health on binge drinking. Journal of Digital Convergence, 16(1), 421-430. doi: 10.14400/JDC.2018.16.1.421.
- Methodology, what are the criteriums of inclusion used to considered the subjects as adolescents? Only the age? In this case was it between 12 and 18 years for all the male and females included in this study? Is it correct or what are the limitations? You should discuss that.
à Thank you for your advice. The youth participating in this study used stratified cluster extraction data by applying proportional allocation to match the population composition ratio and sample composition ratio as a stratified variable for the population aged 12 to 18. The annual survey, conducted by the Korean government to investigate the health behavior of teenagers, has a limitation that non-schoolers are excluded because the survey was conducted by selecting students attending school as a population. However, these are not 4.2% of all teenagers. As we sympathized with the reviewer's opinion, adding the following sentences to the text explained the sampling process according to the population.
“As a middle and high school student nationwide, the extraction framework for sample design used data from middle and high schools nationwide as of April 2018, April 2019, and August 2020. Sample extraction consisted of three stages: population stratification, sample allocation, and sample extraction. In the population stratification stage, local groups and school levels were used as stratification variables. In the sample allocation stage, the proportional allocation method was applied to ensure that the population composition ratio and the sample composition ratio by stratification variable number matched. The stratification extraction method was used for sample extraction, and the first extraction unit was school and the second extraction unit was class.”
- Line 317 check what you written to be clearer for future lecturers.
Sleep quality improve in 2020, in the discussion you should discuss that may be because of the diminution of the circulation, less noise, less pollution, … could have had positive impact especially in the big cities. What about comparation between urban and rural on this topic?
You should discuss why you did not use statistic procedures do control confounding factors or multivariate logistic regression.
à Thank you for your advice. We agree with your suggestions and added the following sentence.
For this data, a proportional allocation method was applied to Korean adolescents aged 12 to 18 attending school so that the population and sample composition ratio match as stratification variables for local groups and school levels. In addition, since samples were taken using the stratified colony extraction method, it is not limited to regions or schools and is generally representative of the characteristics of Korean adolescents. Due to the COVID-19 outbreak in Wuhan in December 2019, Korea, which is close to the region, was relatively quickly under national control. In particular, as adolescents' daily lives changed significantly at the time of the arrival of COVID-19, cross-analysis was used to examine whether the timing was related to the health behavior of various adolescents by setting it as an independent variable rather than other qualitative variables. According to the reviewer's opinion, it is also necessary to analyze the relationship with various health behaviors by setting the region as an independent variable or to derive predictive variables using multiple regression analysis. However, in this study, based on a vast amount of publicly trusted data conducted through a national survey, various social changes caused by COVID-19 were classified into periods to derive differences in health behavior, so the opinions on follow-up studies were added.
“This study is meaningful in that the health behavior of adolescents was investigated nationally, a large amount of reliable data was used, and the research was conducted under the theme of COVID-19, which affected the lives of the population around the world. In addition, it is judged that it is meaningful in that it did not simply compare the data on youth health behavior before the outbreak of COVID-19 (2019) and after the outbreak of COVID-19 (2020), but compared the 2018 data together to confirm whether it is a youth health behavior trend or a change caused by the outbreak of COVID-19.
Meanwhile, this study has the following limitations. First of all, due to the complexly connected social structure, the causal relationship was not derived by control-ling social changes caused by COVID-19 or all disturbance variables within its structure, and secondly, the relationship between physical activity, sleep, obesity, and mental health of adolescents each year was analyzed through cross-analysis, so it is not possible to explain the size or predictive model of health behavior factors caused by COVID-19. Therefore, in follow-up studies, it is necessary to study the health behavior of teenagers by region or to derive a model for predicting health behavior of teenagers in Pandemic situations such as COVID-19.”
- What are the limits of the study, and suggestions for future studies?
à Thank you for your advice. We agree with your suggestions and added the following sentence.
“The result that social isolation and loneliness cause high levels of depression and anxiety in adolescents (Loades ME, Chatburn E, Higson-Sweeney N, Reynolds S, Shafran R, Brigden A, et al, 2020), Home learning and social distancing caused by COVID-19 are the depression of teenagers. Anxiety, Austria's clinical association with sleep and eating disorders (Pieh, Plener, Probst T, Dale R, Humer E, 2021) shows different results on the mental health of adolescents during the COVID-19 period. This study utilizes data collected from early COVID-19 and pre-COVID-19 data, and has limitations in that it is not a study to derive active support or preventive policies for various mental health problems. Therefore, caution is required in the interpretation and application of results, and research to compare various information with longitudinal research needs to be carried out in the future. In particular, in order to manage the mental health of COVID-19 teenagers, mental health problems and various problems that occur in the process of adaptation may occur, so various situations should be considered together.”

Round 2
Reviewer 2 Report
Thank you for revising the manuscript. However, there are some issues that the author cannot clarify. Please see comments.
1. Lines 31-32. “COVID-19 has led to changes …”. This is a very strong causal statement. However, based on the methodology of this study, the authors cannot draw such a conclusion.
2. Lines 115-122. Sampling procedures and details are not clear enough. Authors may consult an epidemiologist or biostatistician regarding relevant procedures. These procedures are the focus of this study.
3. Table 1. According to the purpose of this study, the authors should examine differences in all general characteristics between the years before and after COVID-19, as these differences may affect the results that this study wanted to explored. However, the authors did not correctly analyze the focus of the study.
4. Table 2. The authors responded that “This is the one the series of the manuscripts using the national data set so we firstly examined how adolescent’s health related variables has proportionally changed and the second series of the manuscript that we are preparing should be examined the association between years and variables. In addition, including the association results should take a lot of information and does not fit in one manuscript. Hopefully, you understood.” I disagree with this response. In this study, the results may have been confounded by confounding factors. Therefore, the results in Table 2 do not support the conclusions drawn in this study. If authors have data to clarify the questions, they should do a complete and appropriate analysis to investigate and answer the research questions, rather than leaving some data to put in a second paper.
Author Response
Responses to Editor and Reviewer #2's Comments
- Lines 31-32. “COVID-19 has led to changes …”. This is a very strong causal statement. However, based on the methodology of this study, the authors cannot draw such a conclusion.
REPLY:
Thanks for your suggestions. We agree with your suggestions and the sentence has been revised.
“In 2020, when COVID-19 occurred, it was confirmed that there was a change in the health behavior of adolescents compared to 2018 and 2019. Therefore, active responses from schools, families, and communities are required to foster healthy lifestyle habits in social changes such as COVID-19.”
- Lines 115-122. Sampling procedures and details are not clear enough. Authors may consult an epidemiologist or biostatistician regarding relevant procedures. These procedures are the focus of this study.
REPLY:
Thanks for your comments! We addressed the general sampling procedures and the details information can be found in Materials and Method section. In addition, this national data set collected by the Korea Disease Control and Prevention Agency so we don't think we need to explain in very detail.
- Table 1. According to the purpose of this study, the authors should examine differences in all general characteristics between the years before and after COVID-19, as these differences may affect the results that this study wanted to explored. However, the authors did not correctly analyze the focus of the study.
REPLY:
Thanks for your comments: all general characteristics may should affect the results of the study but the purpose of this study was to examine the proportional difference in the health behavior (i.e., PA, Obesity, Sleep, Mental Health Management), not the association between the variables.
- Table 2. The authors responded that “This is the one the series of the manuscripts using the national data set so we firstly examined how adolescent’s health related variables has proportionally changed and the second series of the manuscript that we are preparing should be examined the association between years and variables. In addition, including the association results should take a lot of information and does not fit in one manuscript. Hopefully, you understood.” I disagree with this response. In this study, the results may have been confounded by confounding factors. Therefore, the results in Table 2 do not support the conclusions drawn in this study. If authors have data to clarify the questions, they should do a complete and appropriate analysis to investigate and answer the research questions, rather than leaving some data to put in a second paper.
REPLY:
Thanks for your comments: We clearly explain or answer the research questions that we intended to do in the manuscript. You addressed that the appropriate analysis should be perform to answer the research questions but the chi-square test should be good enough to examine the proportional differences for the health behavior variables. We understood that it might be better to use multinomial logistic equation to examine the association between the variables but the proportional difference could be performed first before we investigate the further analysis.
